# An Integrated Analysis of Value-Based Adoption Model and Information Systems Success Model for PropTech Service Platform

**Jinmin Kim** and **Jaeyoung Kim** *

Department of Corporate Management, Korea University, Seoul 30019, Korea; tristan1031@korea.ac.kr
* Correspondence: korean4u@korea.ac.kr

**Abstract:** This study develops a sustainable PropTech service model. Specifically, it analyzes the consumer-oriented and service provider-oriented service elements to build a sustainable ecosystem. This study also conducts a preliminary examination to standardize an efficient service platform reflecting the consumer and service factors. Hence, the study analyzes the service factors of users and derives a reference for the quality-of-service design using the value-based adoption model and the information systems success model, respectively. By integrating these models, we develop a tool to analyze empirically the important elements influencing PropTech users. To this end, we conducted a questionnaire survey among 530 PropTech users from 24 August 2021 to 14 September 2021. The results show that the consumer's intention to use services is influenced by service practicability in terms of consumer value. The intention to continue using a service is influenced by user satisfaction. The results also highlight the importance of considering all the quality factors when building a user-friendly service platform. These findings have implications in that they show the importance of a sustainable PropTech service platform in resolving information asymmetry, promoting transparent transactions, and enhancing productivity in the real estate industry.

**Keywords:** IS success model; PropTech; service business ecosystem; service platform; service standardization; VAM

## 1. Introduction

Technology has become indispensable to the survival of businesses, irrespective of their industry domain. Hence, businesses have started tapping into technology-based innovations. This study considers the application of these innovations to the real estate industry, which is also referred to as PropTech, an acronym for property and technology. Specifically, PropTech integrates cutting-edge technologies to provide innovative services to the real estate industry [1]. These PropTech services are manifested through brokerage services using smartphones or the Internet, valuation based on big data and artificial intelligence (AI), and rental management platforms. These services have been gaining attention, owing to their potential in resolving growing information asymmetry, which impedes market efficiency. Particularly, the development of information technology and AI has enhanced data collection and optimized decision making, respectively, thereby enabling the real estate market to overcome information asymmetry [2]. The development of information technology and the opening of the real estate distribution market have also brought about significant changes to the real estate ecosystem. This development has promoted the emergence of young PropTech companies equipped with the latest technologies that have been steering a change in trading methods and business processes [3].

In this context, it must be noted that, similar to other IT-based service platforms, the PropTech industry has been witnessing a wide application of its services. However, the expansion scope of the PropTech industry is wider than that of the other industries. PropTech services are limited to real estate brokerage platforms. However, these services

are expected to expand with the expansion of the real estate ecosystem into domains with much larger individual businesses, such as B2B, construction, and architecture [3]. Concerning the delivery of these services, through the service platform, the platform provider provides a new business model through the new service. This platform also provides the seller and buyer with substantial real estate information and allows for secure transactions at a low price. Concerning information availability, as stated earlier, despite being a socially important and influential industry, the real estate industry suffers from information asymmetry. This problem is pertinent to Korea. Real estate is one of the most valuable wealth-creating sectors in Korea. With real estate attracting social and political attention in Korea, the country must establish a PropTech platform to resolve information asymmetry, secure transaction transparency, and enhance industrial productivity. This development would also require focusing on the standardization of the platform from the standpoint of policy, as well as technically and industrially.

A standardized business ecosystem is being implemented through servitization in various industries, and the PropTech industry is also promoting the creation of sustainable value to stakeholders. Given this, a sustainable business ecosystem should operate on a standardized platform. To establish a standardized platform, it is necessary to define the business accurately, design a suitable business model, and include an upgradable framework. Tian et al. [4] developed a framework to assess business model designs by employing a case study of a retail business-to-business service ecosystem. Faber et al. [5] proposed a collaborative process for building a business ecosystem that could adapt to a gradually changing environment. A systematic approach to the business ecosystem through well-defined processes is emphasized.

In previous research, innovation in the service ecosystem was structured as a systematic process [6], or a systematic approach [7] that allowed consumers and service providers to accept an innovation infrastructure through sustainable innovation was argued. Previous studies on service standardization suggest that the approach of a sustainable service platform is effective [8]. In the study, for the establishment of a service platform on the Internet, it was attempted to understand the behavioral intentions of consumers [9]. Since previous studies have approached this from the point of view of a service provider or a consumer, this study intends to simultaneously identify the needs of consumers and the factors of service providers due to the characteristics of the service platform. In the given context, this study develops a sustainable PropTech service model. To establish a sustainable platform, it analyzes consumer-oriented and service provider-oriented service elements and conducts a preliminary study to standardize the service platform reflecting these consumer and service factors.

This study considers the participants of the business ecosystem: PropTech service providers and users. It considers both sellers and buyers of real estate as the service users. The study also analyzes the quality level of the service design, because it is key to facilitating efficient transactions on a standardized platform. From the users' perspective, this study investigates factors influencing the continuous use of PropTech services. To this end, this study uses the value-based adoption model (VAM) to analyze the service factors of the service users and uses the information systems (IS) success model to derive a reference for service design quality. It integrates these two models to develop a research model to identify empirically the important elements of a sustainable service platform.

## 2. Literature Review and Hypotheses

### 2.1. PropTech

The radical development and diffusion of IT has been engendering changes in many industries, including the highly conservative real estate industry [10]. The scope of technological application to the real estate sector has given rise to the field of PropTech. PropTech and a wide range of IT platforms, such as fintech, have the potential to provide optimized services through AI based on big data [11]. PropTech services have also contributed toward resolving market imbalances arising from information asymmetry. Specifically, data-driven

PropTech has improved inefficient trading practices in the real estate market. Braesemann and Baum [11] ascertained whether PropTech has transformed real estate into a data-driven market and how the data generated using PropTech can be used to create value. Saiz [12] analyzed the economic value of the application of digital and information technology to the real estate industry and suggested a direction for a new business model. Betzeki [13] discussed the general transaction process executed through the PropTech platform in the rental field. Siniak et al. [14] suggested that the institutional practices should be linked with the changes resulting from PropTech's application to the real estate industry, the implications of PropTech for real estate market transparency, and the value of consumers and service providers. In the case of Korea, the size of the PropTech market in 2020 was 0.2% of the total real estate transaction value, but the related companies more than doubled every year. These changes provide evidence of the changes in the real estate industry, though the changes are slower than the changes in other industries.

Despite these multifarious changes, research on PropTech has been limited to the financial or economic aspects. This gap calls for multi-perspective research on the application of information technology to the real estate industry. It also calls for systematic research on the integrated framework of fintech and PropTech. In this context, Starr et al. [15] suggested a systematic progress in the expansion and integration of PropTech. Given that the development of advanced technology has led to changes in the real estate industry, it is important to adopt a systematic approach through the PropTech platform to define and utilize PropTech.

### 2.2. Service Business Ecosystem

In this context, the advancements in information technology further contribute toward business ecosystem modeling. Specifically, the development of business models using various information technologies on a stable platform enables efficient market activities for both service providers and consumers. Uchihira et al. [16] in particular developed Internet of Things (IoT) services using the business ecosystem design method. From a long-term perspective, Hamsa [17] suggested a change in the role of the market and emphasized the productive interaction between service providers and consumers using IT. In this context, it must be noted that the use of radical information technology also enhances mobility. In the context of the mobility as a service (MaaS) business model, Kamargianni and Matyas [18] proposed an MaaS ecosystem to identify each actor and their roles in the MaaS market. An efficient mobility platform can help these actors to increase their market reach. García et al. [19] proposed a business ecosystem providing personalized services to meet the mobility challenges and needs that emerge with the expansion of MaaS.

Concerning the rapid change in the business ecosystem caused by the COVID-19 pandemic, studies focus on implementing measures that can be adopted to address future risks [20]. This scenario calls for establishing a system that can appropriately respond to the PropTech ecosystem and for conducting research on the use of information technology in response to changes.

To build a sustainable service process, it is necessary to study a standardized service platform. In the standardized service process, service standardization provides customers with standards and procedures for using intangible services and thus improves value of the service. It provides cost reduction and operational efficiency to service providers. Hence, although there is a difference in the degree of standardization in each service industry, it is necessary to determine the optimum level of standardization by simultaneously considering customization and standardization [21]. In this context, it must be noted that the delivery of the right proportion of services can provide greater value to customers and thus increase their satisfaction. In this regard, the implementation of a controlled process can improve efficiency. This service standardization can help customers have confidence in the fairness of services. Focusing on the goal of customer satisfaction, every provider should improve their service quality through service standardization [22]. Standardization also provides service quality standards to justify service-related factors and

enables consumers and service providers to embrace a sustainable innovation infrastructure through a systematic approach [23]. These studies are related to the approaches to research methods [24] and research fields [25] in the service domain. This study focuses on the continuous expansion of the PropTech industry. Given this focus, it is necessary to recognize PropTech as a service and explore the factors critical to a sustainable business model of a standardized service platform.

### 2.3. Integration of the VAM and IS Success Model

To build a sustainable service platform, we analyzed users' performance expectations from and intention to continue using the PropTech service platform. By integrating the VAM and IS success model, this study constructs a model considering both consumers and service providers. We took a systematic approach to model design and adopted the VAM to analyze the perceived value of consumers and the IS success model to standardize the design quality for the IS. We used our integrated model to assess the individual impact. Several studies have adopted the VAM and the IS success model. Zhao et al. [26] considered an integrated model to analyze the intention to continue using microblogs through Internet networks. Wang et al. [27] proposed and analyzed a multidimensional model to analyze the success factors of paid mobile learning applications. They used structural equation modeling (SEM) to analyze simultaneously the factors of the VAM and the IS success model. Zhu et al. [28] proposed a model to analyze the factors influencing the users of car-sharing applications in the sharing economy. The previous studies showed that the VAM and the IS success model are used in services utilizing newly emerging information technologies.

As stated earlier, the VAM has been used to analyze new services resulting from the development of information technology. For example, Kim et al. [29] analyzed the value of the introduction of the mobile Internet (M-Internet) as a new information and communication technology (ICT). The study analyzed the consumer's intention to adopt the mobile Internet in the early days of the M-Internet by analyzing the consumer's benefits from and the sacrifice for the new technologies. Roostika [30] performed an individual-level study in which university students adopted new information technologies to analyze the use of mobile technology for university education. By using the VAM and considering the values of the internal and external variables, Seyal et al. [31] studied the perception of adoption of mobile services in the enterprise sector. In a new approach, Kim et al. [32] used the VAM and analyzed consumer values to identify the factors critical to the introduction of new smart home services. The VAM has been combined with research models such as the expectation–confirmation model to identify the continuance intention toward Internet protocol television [33] and online-to-offline services. It has also been used to analyze the intention to continue using an application [34]. The VAM effectively analyzes consumer value for new services. Since it can be systematically utilized in different manners in combination with other models, we used the VAM in this study.

Delone and McLean [35] devised the IS success model. The factors key to the success of an information system are proposed as standardized quality dimensions by simultaneously considering the system, information, and service qualities. These dimensions influence user satisfaction and the intention to use, which consequently impact an individual's perception of the system's impact. This reflects the mediating effect of user satisfaction and the intention to use on personal influence. Concerning the use of this model, since it is easy for service providers to analyze information systems, it has been used to analyze new services launched on IT platforms. For example, using this model, Koo et al. [36] explained the change and development direction of the banking system post the emergence of Internet banking services. Petter and McLean [37] conducted a meta-analysis to analyze individual-level trends to verify the use of the IS success model and prove its effectiveness in the systematic analysis of the service provision platform. This model allows for an examination of the standards of a new platform using the information system. From the perspective of corporate human resources management, Alshibly [38] conducted a system analysis and identified the critical factors for establishing an internal employee education

system. At an individual level, Roky and Al Meriouh [39] analyzed the influence of an information system in the automobile industry and analyzed the standardized service quality level through structural equations. The IS success model mainly allows for the systematic analysis of emerging fields using structural equations. This analysis is evident from the research on service platforms using information systems [40–42]. In this context, it must be noted that the IS success model can be combined with other research models to analyze effective systems. For example, one study provided a framework for analyzing this model in combination with other models such as the technology acceptance model [43]. Another study analyzed the influence on a website by integrating the IS success model with trust factors [44,45].

## 2.4. Hypothesises

The VAM considers the benefits and sacrifices of consumers to analyze their perceived value for new services. Specifically, the perceived value refers to the sum of the benefits the user obtains from the service and the relative sacrifices the user makes. The identification of the sacrificial value can reflect consumers' decisions in real time. Value was analyzed in terms of productivity of consumers who used PropTech services. In order to reflect the service value of PropTech, consumers' benefits and consumer sacrifices were composed as variables to analyze the perception of the cost that consumers had to pay to use the service in preparation for consumer benefits. Referring to the factors used in the research model, variables were selected from previous studies related to the VAM. As for the specific variables, the perceived consumer benefits consisted of usefulness (USEF), enjoyment (ENJ), and ease of use (EAS) [34,46,47], while perceived consumer sacrifice was defined as technicality (TEC) and perceived fee (PERC) [34,47,48], and the following hypotheses were derived to analyze the effect of consumer value on user satisfaction (USER) and intention to use (INT) according to the composition of the VAM model:

**Hypothesis 1 (H1).** *Perceived benefits influence user satisfaction.*

**Hypothesis 1a (H1a).** *Usefulness influences user satisfaction.*

**Hypothesis 1b (H1b).** *Enjoyment influences user satisfaction.*

**Hypothesis 1c (H1c).** *Ease of use influences user satisfaction.*

**Hypothesis 2 (H2).** *Perceived sacrifice influences user satisfaction.*

**Hypothesis 2a (H2a).** *Technicality influences user satisfaction.*

**Hypothesis 2b (H2b).** *The perceived fee influences user satisfaction.*

**Hypothesis 3 (H3).** *Perceived benefits influence the intention to use.*

**Hypothesis 3a (H3a).** *Usefulness influences the intention to use.*

**Hypothesis 3b (H3b).** *Enjoyment influences the intention to use.*

**Hypothesis 3c (H3c).** *Ease of use influences the intention to use.*

**Hypothesis 4 (H4).** *Perceived sacrifice influences the intention to use.*

**Hypothesis 4a (H4a).** *Technicality influences the intention to use.*

**Hypothesis 4b (H4b).** *The perceived fee influences the intention to use.*

Considering the scalability of this model, this study established the following hypotheses focusing on the standardization of the system. The variables used in the IS success model are concerned with the service quality in the analysis of system quality and information quality. System quality (SYS) [44,49,50] deals with the structural standards of a service, information quality (INF) [44,50,51] deals with the standards of the information provided, and service quality (SER) [44,49,51] provides for consumers to measure the standards of the service being provided. To combine the VAM and the IS success model, user satisfaction (USER) and intention to use (INT) were placed together as dependent variables.

**Hypothesis 5 (H5).** *Standardization factors influence user satisfaction.*

**Hypothesis 5a (H5a).** *Information quality influences user satisfaction.*

**Hypothesis 5b (H5b).** *System quality influences user satisfaction.*

**Hypothesis 5c (H5c).** *Service quality influences user satisfaction.*

**Hypothesis 6 (H6).** *Standardization factors influence the intention to use.*

**Hypothesis 6a (H6a).** *Information quality influences the intention to use.*

**Hypothesis 6b (H6b).** *System quality influences the intention to use.*

**Hypothesis 6c (H6c).** *Service quality influences the intention to use.*

Based on the objective to establish a sustainable service platform, this study assessed personal influences in relation to users' continuance of use and performance expectancy. For stable platform operation, it is important to retain frequent service users and to analyze the factors triggering repurchase decisions. Studies in various fields have analyzed major factors influencing users' continuance of use. For example, Ma et al. [52] analyzed the influencing factors by considering the mediating effect of customer satisfaction on the intention to continue overseas shopping. Eugene Cheng-XI et al. [53] and Jiang et al. [54] reported the impact on the sustainability of car-sharing services in the sharing economy. Nascimento et al. [55] and Nguyen et al. [56] presented new services using new technologies such as smart watches and chatbots. These studies analyzed the intention of consumers to continue using the products.

The penetration of the mobile economy into new domains has also motivated similar studies in new areas. For example, one study focused on the influence of service quality on the continuance of use of mobile commerce [57]. Another study analyzed the factors influencing the continuance of use of mobile payment systems [58]. One study considered the development of social media communities and the negative effect of word of mouth on the Internet. Chen et al. [59] studied the intention to continue the use of applications with an extremely high social impact. Another representative study explained the impact of social influence and identity on the intention to continue use [60].

In this study, continuance of use (CON) and performance expectancy (PERF) were selected as the variables for personal influence. The continuance of use is determined by the size of an individual's performance expectancy. The greater the positive impact on individual performance expectancy, the greater the number of loyal users of the service platform will be. This study set the performance expectancy as the second factor of individual impact. Performance expectancy significantly influences an individual's behavioral intention. Do Nam Hung et al. [61] conducted an empirical analysis of users' performance expectations from a mobile payment service system. Lim and Oh [62] examined the effect of performance expectations on clouding services when studying the diffusion of new innovations. Ghalandari [63] studied the influence of performance expectations on mobile banking services, mediated by age and gender. Shaikh et al. [64] reported the effect of

performance expectations and consumer attitudes on the adoption of mobile banking services by considering the perceived risks of adoption. Some studies have analyzed performance expectancy in combination with the unified theory of acceptance and use of technology (UTAUT) model to provide implications for a new service platform [65,66]. Based on this use, this study established the following hypotheses to analyze the major factors influencing the establishment of the PropTech service ecosystem and to derive implications for standardization:

**Hypothesis 7 (H7).** *User satisfaction influences the intention to use.*

**Hypothesis 8 (H8).** *User satisfaction influences individual impact.*

**Hypothesis 8a (H8a).** *User satisfaction influences continuance of use.*

**Hypothesis 8b (H8b).** *User satisfaction influences performance expectancy.*

**Hypothesis 9 (H9).** *Intention to use influences individual impact.*

**Hypothesis 9a (H9a).** *Intention to use influences continuance of use.*

**Hypothesis 9b (H9b).** *Intention to use influences performance expectancy.*

Figure 1 presents the proposed conceptual model and nine hypothesized relationships. Figure 1 also represents the integrated conceptual model constructed to assesses the relationships between the constructs. We adapted the relationships between the constructs from the related literature.

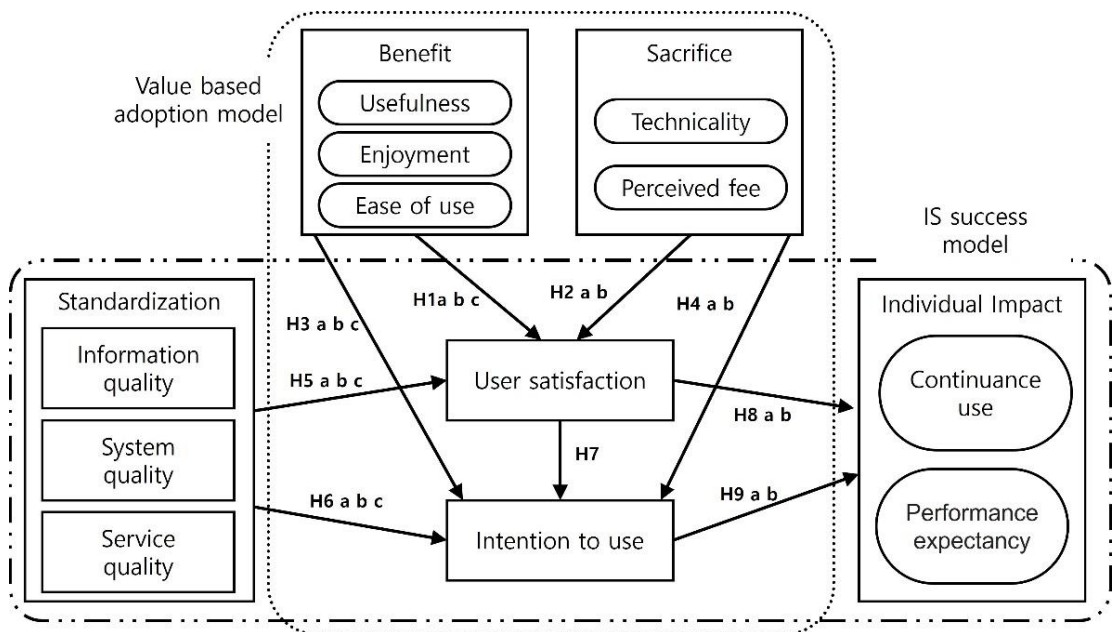

**Figure 1.** Research model.

## 3. Research Methodology

Currently, several PropTech service platforms including Dabang (https://www.zigbang.com (accessed on 15 November 2021)) and Jigbang (https://www.dabangapp.com (accessed on 15 November 2021)) are providing services in Korea. In this study, to measure the value of the service experience and the quality of the service platform, we used the conceptual model to develop a research tool for empirically analyzing the PropTech service.

The respondents were consumers who used a service platform that provided Proptech business based on the Internet at least once. We conducted a mobile questionnaire survey from 24 August 2021 to 14 September 2021 and registered 723 responses. After screening the responses, we obtained a valid sample of 530 respondents and a response rate of 73.31%. Table 1 lists the demographic information of these respondents. The PLS-SEM results confirmed that 530 respondents met the minimum sample size requirement for analysis [67]. Hair et al. [67] provided a table that shows the minimum sample size requirement necessary to detect the minimum $R^2$ values in the maximum number of arrows pointing at a construct of any significance level.

**Table 1.** Demographic information.

| Respondent's Demographic Profile | Variables | Usable Responses | Percentage (%) |
|---|---|---|---|
| Gender | Male | 196 | 36.98 |
| | Female | 334 | 63.02 |
| Age | Below 19 | 10 | 1.89 |
| | 19–35 | 246 | 46.42 |
| | 35–50 | 225 | 42.45 |
| | Above 50 | 49 | 9.25 |
| Education qualification | Undergraduate | 61 | 11.51 |
| | Graduate | 406 | 76.60 |
| | Postgraduate | 63 | 11.89 |
| Household income (in a month, USD) | Below 1000 | 66 | 12.45 |
| | 1000~2000 | 75 | 14.15 |
| | 2000~3000 | 189 | 35.66 |
| | 3000~4000 | 93 | 17.55 |
| | 4000~5000 | 53 | 10.00 |
| | Above 5000 | 54 | 10.19 |

To provide accurate responses and reduce ambiguity, we conducted a pilot study with 50 members of academia and experts handling PropTech services. The pilot group filled the questionnaire correctly and suggested a slight change in the language of the questionnaire. After incorporating the proposed modifications, we obtained a final questionnaire with closed-ended questions answered on a 7-point Likert scale. The questions were appropriate, given that they were based on the responses provided by experienced PropTech services' users. The items of the research tools were based on a literature review relevant to the research model. Some texts were supplemented and modified to fit the context of the PropTech services. Specifically, we attempted to refine the tool by removing items with low adjusted item total correlations and by performing an exploratory factor analysis to drop items that were not properly loaded into the factors.

The final questionnaire in Table 2 comprised the following items: CON (6 items), EAS (5 items), ENJ (3 items), INF (5 items), INT (3 items), PERC (3 items), PERF (3 items), SER (5 items), SYS (5 items), TEC (3 items), USF (8 items), and USER (4 items). Table 2 shows the final measurement instrument used and their sources of operationalization. After explaining the characteristics of the respondents using descriptive statistics, we analyzed the questionnaires using the PLS-SEM technique, a multivariate analysis technique that has recently gained academic attention. PLS-SEM is a powerful tool that exerts minimal restrictions on the measurement scale and is useful for modeling latent constructs under non-normal conditions. When conducting PLS-SEM, it was necessary to perform non-iterative applications of the general least squares regression to ensure the reliability and validity of the measurement model and to obtain the external weights, loads, and structural model relationships for the latent and manifest variables. Finally, this study applied a bootstrap resampling procedure to evaluate the statistical significance of the structural pathways. Before the PLS-SEM analysis, we coded the data using the statistical package for the social sciences 20 for statistical and correlation analyses. We used a complete SEM tool, SMARTPLS 3.0, to test the hypothesis. Since this study is an exploratory study to

analyze the intrinsic latent variables of the PropTech service, PLS-SEM was used rather than CB-SEM, which is suitable for the verification of the theory [68].

**Table 2.** Questionnaire source and number of items.

| Constructs | Number of Items | Sources |
|---|---|---|
| Usefulness (USEF) | 8 | [34,46,47] |
| Enjoyment (ENJ) | 3 | [34,46,47] |
| Ease of use (EAS) | 5 | [34,46,47] |
| Technicality (TEC) | 3 | [34,47,48] |
| Perceived fee (PERC) | 3 | [34,47,48] |
| Information quality (INF) | 5 | [44,50,51] |
| System quality (SYS) | 5 | [44,49,50] |
| Service quality (SER) | 5 | [44,49,51] |
| User satisfaction (USER) | 4 | [46–48,50] |
| Intention to use (INT) | 3 | [46,48–50] |
| Continuance of use (CON) | 6 | [61,62,64] |
| Performance expectancy (PERF) | 3 | [61,62,64] |

## 4. Results

To test the model, this study drew on SmartPLS 3.0 with the path weighting scheme. The bootstrapping procedure drew 125 cases and 5000 samples using the no sign change option. When evaluating and reporting the results, we followed the recent PLS-SEM guidelines given by, for example, Hair et al. [67,69] and assessed the measurement models before evaluating the structural model.

SmartPLS uses the SRMR and GOF to evaluate the model fit. GOF is a value obtained by multiplying the average value of $R^2$ by the average value of AVE and then taking the square root again. The GOF value of this study model is 0.596, which can be said to have a high goodness of fit [70,71]. The SRMR value is a number calculated based on standardized residuals [72]. If the model fit is perfect, the SRMR becomes 0, and if it is less than 0.08, the model fit is judged to be good. The SRMR of this research model was 0.053, which could be judged to have a high degree of fit. In addition, the RMS_theta value of 0.114 indicates a well-fitting model, and a higher value indicates a poor fit [73].

Table 3 presents the result of the reliability and definitive factor analysis. In general, if the standard loading value was 0.5 or more, then the item could be considered valid. If the mean variance extraction value (AVE) was also 0.5 or more, then the grouping factors could be considered valid [69] composite reliabilities for the five reflectively measured constructs ranging from 0.93 to 0.96, exceeding the minimum requirement of 0.70.

**Table 3.** Validity and reliability of measures.

| Measures | Item | Loading or Weights | Cronbach's Alpha | Composite Reliability | AVE |
|---|---|---|---|---|---|
| Continuance of use (CON) | CON1 | 0.886 | 0.948 | 0.959 | 0.794 |
| | CON2 | 0.884 | | | |
| | CON3 | 0.907 | | | |
| | CON4 | 0.877 | | | |
| | CON5 | 0.903 | | | |
| | CON6 | 0.890 | | | |
| Ease of use (EAS) | EAS1 | 0.918 | 0.944 | 0.958 | 0.819 |
| | EAS2 | 0.911 | | | |
| | EAS3 | 0.923 | | | |
| | EAS4 | 0.922 | | | |
| | EAS5 | 0.847 | | | |
| Enjoyment (ENJ) | ENJ1 | 0.907 | 0.885 | 0.928 | 0.812 |
| | ENJ2 | 0.924 | | | |
| | ENJ3 | 0.872 | | | |

**Table 3.** *Cont.*

| Measures | Item | Loading or Weights | Cronbach's Alpha | Composite Reliability | AVE |
|---|---|---|---|---|---|
| Information quality (INF) | INF1 | 0.790 | | | |
| | INF2 | 0.877 | | | |
| | INF3 | 0.885 | 0.91 | 0.933 | 0.737 |
| | INF4 | 0.866 | | | |
| | INF5 | 0.872 | | | |
| Intention to use (INT) | INT1 | 0.941 | | | |
| | INT2 | 0.933 | 0.93 | 0.955 | 0.877 |
| | INT3 | 0.934 | | | |
| Perceived fee (PERC) | PERC1 | 0.891 | | | |
| | PERC2 | 0.895 | 0.777 | 0.872 | 0.697 |
| | PERC3 | 0.704 | | | |
| Performance expectancy (PERF) | PERF1 | 0.895 | | | |
| | PERF2 | 0.927 | 0.896 | 0.935 | 0.828 |
| | PERF3 | 0.907 | | | |
| Service quality (SER) | SER1 | 0.867 | | | |
| | SER2 | 0.850 | | | |
| | SER3 | 0.876 | 0.924 | 0.942 | 0.766 |
| | SER4 | 0.891 | | | |
| | SER5 | 0.892 | | | |
| System quality (SYS) | SYS1 | 0.889 | | | |
| | SYS2 | 0.886 | | | |
| | SYS3 | 0.881 | 0.911 | 0.934 | 0.74 |
| | SYS4 | 0.900 | | | |
| | SYS5 | 0.733 | | | |
| Technicality (TEC) | TEC1 | 0.951 | | | |
| | TEC2 | 0.913 | 0.933 | 0.954 | 0.874 |
| | TEC3 | 0.939 | | | |
| Usefulness (USEF) | USEF1 | 0.875 | | | |
| | USEF2 | 0.870 | | | |
| | USEF3 | 0.890 | | | |
| | USEF4 | 0.884 | 0.952 | 0.960 | 0.75 |
| | USEF5 | 0.815 | | | |
| | USEF6 | 0.850 | | | |
| | USEF7 | 0.857 | | | |
| | USEF8 | 0.886 | | | |
| User satisfaction (USER) | USER1 | 0.878 | | | |
| | USER2 | 0.902 | 0.919 | 0.943 | 0.805 |
| | USER3 | 0.923 | | | |
| | USER4 | 0.885 | | | |

In this study, to examine the common method variance (CMV) that may occur in PLS-SEM, the VIF among the potential factors (proposed by Knock [74]) was checked. As a result of confirming multicollinearity in the path between the latent variables, the VIF showed a minimum value of 1.262 and a maximum value of 2.917, which did not exceed the threshold of 5. CMV was not an issue in this study. In addition, since the correlation coefficient between the variables was not large, the possibility of CMV was judged to be small [75].

The Fornell and Larcker [76] criterion demonstrated that all AVE values for the reflective constructs were higher than the squared interconstruct correlations, thereby indicating discriminant validity. Similarly, all indicator loadings were higher than their respective cross loadings, providing further evidence of discriminant validity. Table 4 shows the AVE values on the diagonal and the squared interconstruct correlations off the diagonal.

**Table 4.** Discriminant validity results.

| □ | CON | EAS | ENJ | INF | INT | PERC | PERF | SER | SYS | TEC | USEF | USER |
|---|---|---|---|---|---|---|---|---|---|---|---|---|
| CON | 0.891 | □ | □ | □ | □ | □ | □ | □ | □ | □ | □ | □ |
| EAS | 0.660 | 0.905 | □ | □ | □ | □ | □ | □ | □ | □ | □ | □ |
| ENJ | 0.662 | 0.609 | 0.901 | □ | □ | □ | □ | □ | □ | □ | □ | □ |
| INF | 0.673 | 0.656 | 0.699 | 0.859 | □ | □ | □ | □ | □ | □ | □ | □ |
| INT | 0.804 | 0.686 | 0.627 | 0.656 | 0.936 | □ | □ | □ | □ | □ | □ | □ |
| PERC | 0.157 | 0.076 | 0.278 | 0.249 | 0.161 | 0.835 | □ | □ | □ | □ | □ | □ |
| PERF | 0.810 | 0.703 | 0.641 | 0.713 | 0.801 | 0.195 | 0.910 | □ | □ | □ | □ | □ |
| SER | 0.719 | 0.706 | 0.720 | 0.851 | 0.720 | 0.216 | 0.749 | 0.875 | □ | □ | □ | □ |
| SYS | 0.692 | 0.676 | 0.630 | 0.824 | 0.685 | 0.194 | 0.726 | 0.827 | 0.860 | □ | □ | □ |
| TEC | 0.032 | −0.196 | 0.191 | 0.123 | −0.024 | 0.638 | −0.022 | 0.085 | 0.060 | 0.935 | □ | □ |
| USEF | 0.778 | 0.786 | 0.699 | 0.754 | 0.788 | 0.162 | 0.819 | 0.802 | 0.760 | −0.032 | 0.866 | □ |
| USER | 0.811 | 0.685 | 0.696 | 0.743 | 0.791 | 0.228 | 0.861 | 0.758 | 0.716 | 0.074 | 0.781 | 0.897 |

To confirm discriminant validity, the heterotrait/monotrait ratio of correlations (HTMT) was also evaluated as suggested by Henseler et al. [77] (Table 5). If the HTMT value was lower than 0.90, it was evaluated that discriminant validity was secured. In this study, it was found to be between 0.042 and 0.847, reaffirming the security of discriminant validity.

**Table 5.** Heterotrait/monotrait ratio of correlations.

| □ | CON | EAS | ENJ | INF | INT | PERC | PERF | SER | SYS | TEC | USEF | USER |
|---|---|---|---|---|---|---|---|---|---|---|---|---|
| CON | | | | | | | | | | | | |
| EAS | 0.596 | □ | □ | □ | □ | □ | □ | □ | □ | □ | □ | |
| ENJ | 0.614 | 0.558 | □ | □ | □ | □ | □ | □ | □ | □ | □ | |
| INF | 0.623 | 0.605 | 0.672 | □ | □ | □ | □ | □ | □ | □ | □ | |
| INT | 0.777 | 0.663 | 0.608 | 0.687 | □ | □ | □ | □ | □ | □ | □ | |
| PERC | 0.754 | 0.631 | 0.582 | 0.611 | 0.777 | □ | □ | □ | □ | □ | □ | |
| PERF | 0.081 | 0.093 | 0.237 | 0.195 | 0.086 | 0.135 | □ | □ | □ | □ | □ | |
| SER | 0.668 | 0.655 | 0.689 | 0.827 | 0.676 | 0.722 | 0.156 | □ | □ | □ | □ | |
| SYS | 0.642 | 0.628 | 0.597 | 0.804 | 0.64 | 0.701 | 0.136 | 0.802 | □ | □ | □ | |
| TEC | 0.085 | 0.111 | 0.114 | 0.069 | 0.042 | 0.047 | 0.636 | 0.085 | 0.098 | □ | □ | |
| USEF | 0.715 | 0.729 | 0.649 | 0.706 | 0.736 | 0.786 | 0.091 | 0.753 | 0.711 | 0.085 | □ | |
| USER | 0.767 | 0.635 | 0.664 | 0.711 | 0.755 | 0.847 | 0.166 | 0.722 | 0.68 | 0.079 | 0.732 | |

The structural model for the results is shown in Figure 2. The R-squares are also provided for judging the path coefficients of the endogenous latent variables. Most of the path coefficients with significance were found to be related at a level of $p \leq 0.01$. The path coefficient of $p \leq 0.05$ (ease of use -> user satisfaction and information quality -> intention to use) and the path coefficient of $p \leq 0.10$ (system quality -> intention to use and service quality -> intention to use) shoed a statistical relationship and indicated that meaningful analysis was possible. Table 6 shows all the calculated values.

Unlike previous studies, some path coefficients did not appear to be statistically significant. In the case of the VAM, enjoyment and ease of use of the perceived benefit did not affect the intention to use, and both technicality of the perceived sacrifice and perceived fees did not affect user satisfaction or the intention to use. In the IS success model, the information quality and system quality did not affect user satisfaction. The reason why they did not have a significant impact was considered to reflect the characteristics of the PropTech service and will be discussed in the next chapter.

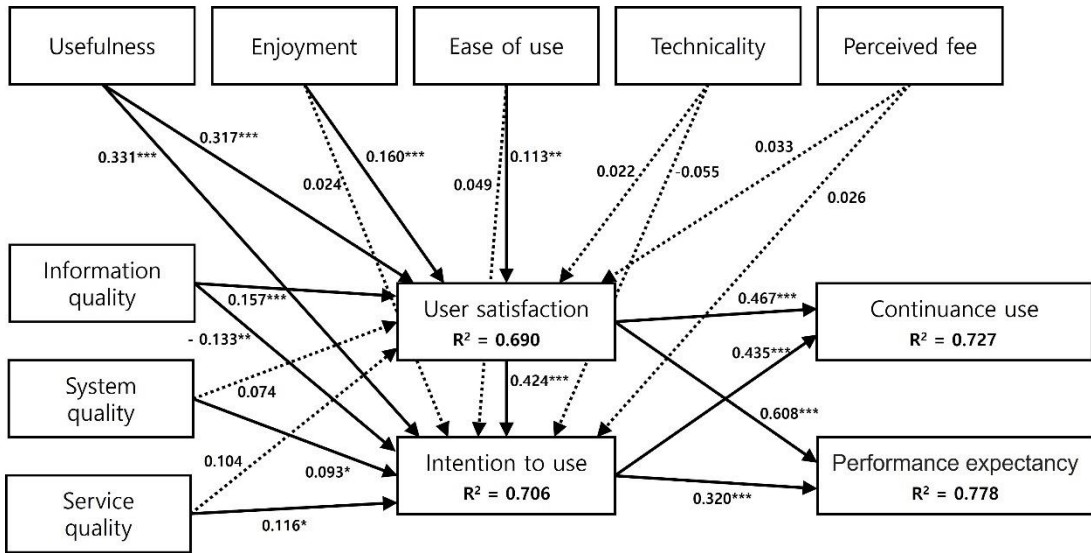

**Figure 2.** Structural equation model. Notes: *** $p \leq 0.01$; ** $p \leq 0.05$; and * $p \leq 0.10$. Dashed lines represent non-significant relationships.

**Table 6.** Hypothesis testing results.

| Hypothesis | | Relationship | Total Effect | T-Value | *p*-Value | Status |
|---|---|---|---|---|---|---|
| H1 | a | Usefulness -> User Satisfaction | 0.317 | 6.036 | 0 | Accept |
| | b | Enjoyment -> User Satisfaction | 0.16 | 3.859 | 0 | Accept |
| | c | Ease of Use -> User Satisfaction | 0.113 | 2.337 | 0.02 | Accept |
| H2 | a | Technicality -> User Satisfaction | 0.022 | 0.476 | 0.635 | Reject |
| | b | Perceived Fee -> User Satisfaction | 0.033 | 0.871 | 0.384 | Reject |
| H3 | a | Usefulness -> Intention to Use | 0.331 | 6.092 | 0 | Accept |
| | b | Enjoyment -> Intention to Use | 0.024 | 0.486 | 0.627 | Reject |
| | c | Ease of Use -> Intention to Use | 0.049 | 0.982 | 0.326 | Reject |
| H4 | a | Technicality -> Intention to Use | −0.055 | 1.255 | 0.21 | Reject |
| | b | Perceived Fee -> Intention to Use | 0.026 | 0.673 | 0.501 | Reject |
| H5 | a | Information Quality -> User Satisfaction | 0.157 | 2.696 | 0.007 | Accept |
| | b | System Quality -> User Satisfaction | 0.074 | 1.36 | 0.174 | Reject |
| | c | Service Quality -> User Satisfaction | 0.104 | 1.627 | 0.104 | Reject |
| H6 | a | Information Quality -> Intention to Use | −0.133 | 2.277 | 0.023 | Accept |
| | b | System Quality -> Intention to Use | 0.093 | 1.763 | 0.079 | Accept |
| | c | Service Quality -> Intention to Use | 0.116 | 1.746 | 0.081 | Accept |
| H7 | | User Satisfaction -> Intention to Use | 0.424 | 7.941 | 0 | Accept |
| H8 | a | User Satisfaction -> Continuance Use | 0.467 | 10.539 | 0 | Accept |
| | b | User Satisfaction -> Performance Expectancy | 0.608 | 15.458 | 0 | Accept |
| H9 | a | Intention to use -> Continuance Use | 0.435 | 9.625 | 0 | Accept |
| | b | Intention to use -> Performance Expectancy | 0.32 | 7.639 | 0 | Accept |

The influence of consumer value shows that the perceived benefits (usefulness, enjoyment, and ease of use) had a positive effect on user satisfaction. Owing to this statistical significance, H1 was accepted. Specifically, usefulness (0.317) showed the largest path coefficient, and only usefulness (0.331) showed a positive relationship with the intention to use. However, both H2 and H4 were rejected, as neither consumer satisfaction nor perceived sacrifice influenced user satisfaction or the intention to use.

Concerning the impact of the service platform quality, among the information, system, and service qualities, only information quality influenced user satisfaction, and hence H5 was partially accepted. However, the information (−0.133), system (0.093), and service qualities (0.116) influenced the intention to use, and hence H6 was accepted. The analysis of the impact on personal impact showed that user satisfaction positively influenced the



intention to use (0.424), and user satisfaction and intention to use influenced continued use (0.467) and performance expectation (0.608). Hence, H7, H8, and H9 were accepted.

## 5. Conclusions

This is a preliminary study on the standardization of a sustainable PropTech service platform, which is based on an integrated model comprising the VAM and the IS success model. While the VAM was used to analyze the consumer values comprising perceived consumer benefits and sacrifices, the quality of the service platform was analyzed through the IS success model.

We derived the following results. As aspects of theoretical implications, first, for the perceived consumer benefits, it was analyzed that consumer benefits (usefulness, enjoyment, and ease of use) all had a positive effect on user satisfaction. In particular, usefulness (0.317) showed the largest path coefficient on user satisfaction, and only usefulness (0.331) showed a positive relationship with the intention to use. This reflects that customers are satisfied when the benefits return to them. However, since only usefulness affects the intention of use of PropTech, consumers' use of the service is practical. Considering the importance of the PropTech market, this reflects the intention to use the service, emphasizing practicality rather than simple interest and convenience. In the case of perceived consumer sacrifice, neither technicality nor perceived fees appeared to affect user satisfaction or the intention to use. Vishwakarma et al. [78] showed a negative influence, and Wang et al. [79] did not show a significant effect on the technicality or perceived fees of consumer sacrifice, and this was due to the characteristics of each service field. Thus, further research and extensive analysis are required. Second, in the IS success model, it appeared that only information quality affected user satisfaction. Regarding the information asymmetry that occurs the most in the real estate market, it was proven that it is important for consumers to provide sufficient information. It was shown that information quality (−0.133), system quality (0.093), and service quality (0.116) affected the intention to use. In particular, the information quality showed a negative (−) value, which was analyzed to be because the quality of the currently provided information was remarkably low. Third, user satisfaction and the intention to use had an effect on the continued use and performance expectations. The significant influence of the path coefficients supports the results of previous theories and studies, and this is also applied to PropTech services.

As for the aspects of practical implications, first, concerning consumer value for PropTech services, this study showed that the intention to use the service was determined by service practicability rather than interest and convenience. The PropTech service users indicated their ability to afford or the effort to use the PropTech service. This implies that the degree of sacrifice does not influence their satisfaction or intention to use it. In other words, it can be interpreted that it is important to achieve the purpose of transaction (transaction of real estate), regardless of the sacrifice. Considering the characteristics of PropTech services, the results show the importance of providing more satisfactory services that can motivate users to pay for or make an effort to use the technology. Second, in order to build a service platform meeting the required quality levels, it is necessary to build a consumer-friendly platform by considering all the quality factors. Particularly, the results of this study showed that the users considered the quality of the currently provided information to be remarkably low. This shows that consumers demand high-quality information, which implies the need for sufficient information about the service through the establishment of a service platform. Third, in terms of personal impact, it supports the previous research showing that the intention to continue use depends on user satisfaction [80]. This means that high-quality PropTech services can lead to the sustainability of the business platform. Users expect to receive useful value through the PropTech service. This finding shows the importance of consumer value as an asset in the real estate market. It also implies the strong public character for basic residential space. In the future, PropTech users would need a strategic approach to understand real estate as an

asset value or to provide customized services by analyzing consumers' intention to use PropTech by dividing it into an approach to secure a basic residential space.

This study has an important academic contribution in that it selects PropTech as the main research subject, which has received little academic attention. In the meantime, Fintech, cryptocurrency, and robo-advisor have been mainly attracting attention given their importance to the Fourth Industrial Revolution or to the development of financial markets and ICT. To the best of our knowledge, this study on the standardization of a sustainable service platform has received scanty research attention. This study is a call to examine this topic further. In addition, this study integrated and presented the existing theories through empirical research, and from the results, it suggests the need for research to build a continuous service platform. Increasing value through servitization provides sustainable business opportunities.

From a policy perspective, the introduction of PropTech and structural innovation have brought about a fundamental change in the policy perspective that tries to interpret real estate investment from the perspective of speculation. This technology has also resolved information asymmetry between consumers and service providers. This factor increases market instability and hinders market efficiency. Hence, this study intends to propose the spread of PropTech-based services as a method to correct such information asymmetry and to find practical implications by presenting a sustainable business model for future PropTech businesses.

The limitations of this study and the scope for future studies are as follows. First, it is necessary to think beyond the asset value of real estate and the basic consumer value. Consumer service use should be classified according to the purpose of the transaction. It is also necessary to conduct structural research on the differentiated provision of service platforms according to each consumer's intention to use. Second, there is a vast range of products according to the value (i.e., prices) of real estate products. This aspect has not been addressed in this study. Hence, it is necessary to conduct a more detailed analysis according to product classification. This business ecosystem aims to provide optimized product information using big data obtained through the combination of the latest information technologies. This aspect can also motivate future research. The future studies can also focus on the optimization service of AI based on value analysis or the provision of non-face-to-face services through virtual or augmented reality. In addition, research is needed to derive a sustainable service strategy in the service ecosystem, where various services are emerging through the rapid development of IT. A theoretical approach suitable for new services is needed, along with research to apply various existing theories to new services. Unlike previous studies, the results of this study, which did not show significant path coefficients, mean that appropriate variables should be found by analyzing the characteristics of each service.

**Author Contributions:** Methodology, J.K. (Jaeyoung Kim); Project administration, J.K. (Jaeyoung Kim); Writing—original draft, J.K. (Jinmin Kim); Writing—review & editing, J.K. (Jaeyoung Kim). All authors have read and agreed to the published version of the manuscript.

**Funding:** This research was supported by the BK21 FOUR (Fostering Outstanding Universities for Research) funded by the Ministry of Education (MOE, Korea) and National Research Foundation of Korea (NRF).

**Data Availability Statement:** Not applicable.

**Conflicts of Interest:** The authors declare no conflict of interest.

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
