# Peer review of "An Integrated Analysis of Value-Based Adoption Model and Information Systems Success Model for PropTech Service Platform"

_sustainability, doi:10.3390/su132312974_

Round 1

Reviewer 1 Report

Reviewer’s Comments

I think that the aim and methodology of this paper are relevant to Sustainability. Especially this is an important theme regarding "Sustainable Management and Business Ecosystems" nowadays. However, there remain issues that need to be addressed.

  1. About Section 1. Introduction, although the author made a good explanation for the introduction. But the reviewer cannot see the value of this research from it. For example, based on the related research in the past to see those shortcomings (whether there are similar studies in the past), or what problems are expected to be solved by introducing this model. What phenomenon can it explain? It is suggested that the author can rethink and rewrite.

  1. About Section 1. Introduction, on page 2, lines 74-80. " This study is organized as follows. Chapter 2 explains the literature that provides a basis for deriving a sustainable business platform. Based on the VAM and IS success model, Chapter 3 draws hypotheses considering the consumers and service providers. Chapter 4 presents the research model, methods, and contents. Chapter 5 analyzes the research results using the partial least squares structural equation modeling (PLS-SEM). Chapter 6 draws conclusions and discusses the limitations and the scope for further research." This paragraph of description could be removed, no more explanation is needed, which can make the introduction more concise and readable.

  1. About Section 2.2. Service Business Ecosystem, on page 3, lines 108-137. The reviewer suggested that the author may consider incorporating this paragraph into the introduction for the explanation. It can show why this is the purpose of research.

  1. About Section 2.3. Service standardization, on pages 3 and 4, lines 138-156. In this paragraph, the author has explained the choice of standardization. However, the literature discussion here does not present the variables to be measured in this dimension. It is suggested that the author(s) should rewrite it and explain who uses the theory proposed by the researcher. Please try to apply it to our service standardization. (EX: Information Quality, System Quality, and Service Quality.)

It is recommended to rewrite the title and content of this section. Author(s) can also refer to the recommendations at the end of the article.

  1. About Section 3.1. Service standardization, on page 4, line 173. The title is different from the content of this paragraph. It is suggested that authors can put the relevance of their research variables in the title for the presentation.

On page 5, line 199. Hypothesis 2. " Perceived satisfaction influences user satisfaction ". The author should confirm the hypothesis because it is different from the research model shown in Figure 1 on page 7.

  1. About Section 3.2. IS success model, on page 5, lines 202-231. The title is partly related to the content, but it does not allow readers to clearly understand the content of this paragraph from the title. The reviewer suggested that this paragraph should be rewritten. And move part of the content to section 2.x for presentation.

  1. About Section 3.3. Individual impact, on page 6, line 232-270. The title is partly related to the content, but it does not allow readers to clearly understand the content of this paragraph from the title. The reviewer suggested that this paragraph should be rewritten. And move part of the content to section 2.x for presentation.

  1. Based on the 3-7 points mentioned above, the reviewers sincerely suggest that the relevant titles and content should be rewritten, and the literature and hypothetical inference methods can be adjusted in a timely manner. Because the current content presentation order will make it difficult for readers to understand.

  1. About Section 3. Hypothesis. Regarding the research model of Hypothesis 1~6 and Figure 1, it is recommended that the author try to use (a), (b), (c), and the research model to indicate the details of each hypothesis (a), (b), (c). On the one hand, it can make the hypothesis more clearly presented, and on the other hand, it can also allow readers to compare the results (if you do this, please add notes (a), (b), (c) to Table 4). Please refer to the picture below.

  1. On pages 7 and 8, lines 277-311. About Section 4. Research Methodology. It is recommended to provide a description of the research object and related PropTech service platform. (Please indicate whether the source of the survey is one or more platforms in the same industry, and what related information services the platform provides. Is it provided by the government or private enterprises? ...etc.)

This information can help readers understand the research objects and sample characteristics more clearly.

  1. On pages 7 and 8, lines 277-311. About Section 4. Research Methodology. It is recommended to provide the dimensions of the questionnaire design and whether there is an explanation of the source of reference to the relevant literature.

  1. On page 8, line 312. About Section 5. Results. Suggest the author(s) should add this paragraph to explain the common-method variance (CMV) of this research. That would be better let readers to understand whether the CMV's situation in this study is not serious.

  1. On page 10, About the analysis of discriminant validity, the reviewer recommends using the HTMT criterion to assess discriminant validity. The authors could reference the below literature.

Henseler, J., Ringle, C.M. & Sarstedt, M. A new criterion for assessing discriminant validity in variance-based structural equation modeling. J. of the Acad. Mark. Sci. 43, 115–135 (2015).

 https://doi.org/10.1007/s11747-014-0403-8

  1. About Section 5. Results. The reviewer knows that this study applied SmartPLS 3.0 to analyze the research model. But finally, it does not provide about Model Fit. Suggest the authors could try to show the Goodness of Fit (GOF) and Standardized Root Mean Square Residual (SRMR). Or

And SmartPLS 3.0 offers the following fit measures:

  • SRMR
  • Exact fit criteria d_ULS and d_G
  • NFI
  • Chi²
  • RMS_theta

  1. About Section 5. Result, on pages 11-12, lines 333-358. However, according to most studies, it is pointed out that the relevant hypothesis is a positive influence, and some of the results of the study appear to be inconsistent with the hypothesis. Suggest the authors should explain the result in more detail. And could try to find some study to extend and illustrate explicitly not support reason. It would be better to let the reader know why to happen the situation.

  1. The author explains the conclusion very clearly. However, the limitations of the research and future research directions are still insufficient, and it is suggested that additional explanations can be made. 

Finally, for the content of this manuscript, the reviewers believe that the concept of standardization is very good. However, the presentation of the article cannot resonate much with readers. The reason is that the performance of the research content cannot make people confirm that your research object, platform, and industry are already the result of standardization (what is the standardization of information technology and platform, who customized). The standardization mentioned in the content can cause readers to misunderstand. It is recommended that authors not only consider modifying terminology standardization (including titles, literature discussions, etc., authors can try to use other words to present the research) but also make substantial changes to the relevant content of the article.

Author Response

We thank you for the feedback on our manuscript, " An integrated analysis of VAM and information systems (IS) success model for PropTech service platform." We did our best to address every concern in the feedback. The review process significantly improved the quality of our manuscript. We really appreciate it. There are several major changes made in the revised manuscript.

It should be noted that we responded to every point from both reviewers, attempted to address it in the revision process, and incorporated it in the revised manuscript, which made the current manuscript somewhat longer than the previous one. If there are any immediate questions or comments, please do let us know and we will be happy to address them.

This study focuses on analyzing critical factors of a sustainable PropTtech service platform. It extends the existing literature by focusing on the PropTech service design and sustainability, standardization of the service quality level, service business ecosystem perspective, consumer and service-oriented service elements in service platform design, influence of factors on consumer value, and an integrated VAM and IS success model.

Further, we believe that this paper will be of interest to the readership of your journal because this study has important implications in that it shows the importance of a sustainable PropTtech service platform in resolving information asymmetry, promoting transparent transactions, and enhancing productivity in the real estate industry. It also shows the influence of PropTtech on policy formulation.

This manuscript has not been published or presented elsewhere in part or in entirety and is not under consideration by another journal. All study participants provided informed consent, and the study design was approved by the appropriate ethics review board. We have read and understood your journal’s policies, and we believe that neither the manuscript nor the study violates any of these. There are no conflicts of interest to declare.

Thank you for your consideration. I look forward to hearing from you.

Sincerely,

Reviewer 2 Report

This manuscript has a very good potential as the focus is on the tenderization of a sustainable PropTech service platform based on an integrated model. It also built a user-friendly platform as a tool to investigate empirically success models. This study will have an important academic contribution because of the selected PropTech as a research subject. Overall it is a good, well-written article, I strongly accepted

Author Response

(The authors gave the same response as above.)

Reviewer 3 Report

  1. There are numerous places where are missing citations. Please add citations throughout the entire paper.
  2. This current study’s background and objectives were weakly developed. They are not enough to provide clear justification for conducting this study. So I highly recommend that the introduction be reorganized to provide a clear flow into the study to the reader by specifically focusing on “what are the key issue?,” “Where is the gap between the literature?,” It would also be valuable to clearly outline more specific potential and meaningful implications that could be delivered for applying into the Proptech context.
  3. In the “PropTech” section, I would like to know real situations of how much Proptech has been applied in the industry? How about users (both buyers and sellers) and their perceptions and satisfaction?? Please add some statistical industry data in terms of Proptech
  4. Lines 158-172, the authors introduced prior literature applying both the VAM and IS success model, however, they did not discuss what those studies revealed, which could have been greatly helpful for readers to understand what new knowledge the authors provided by conducting this study.
  5. The statements of hypotheses do not indicate any specific industry or context. We have already learned enough about the significant links between perceived benefits, sacrifices, satisfaction ad intent to use. Why did we need to learn the same relationships ?? Please specify the study context in the hypotheses.
  6. Please add more explanations regarding sub-dimensions of perceive benefits and sacrificed as presented in Figure 1.
  7. The same comment for Hypotheses 5 and 6
  8. The construct of ‘standardization’ includes three sub-dimensions (information quality, system quality and service quality). This should have been discussed in developing Hypotheses 5 and 6.
  9. Please provide more detailed information on data collection and respondents’ qualification for participation. Who were the study’ samples?? How were respondents contacted?
  10. What criteria were used to screen the responses??
  11. PLS-SEM was used. Why not CB-SEM??
  12. Please what CON, EAS,… etc. stand for?? Spell them out when they first appear
  13. Please prove the appropriateness of the SEM fit. 
  14. Conclusion: Please separate this section into two: theoretical and practical implications. Also provide in-depth discussions by comparing the findings offered by this current study with prior relevant literature’s findings; and also by focusing on what we do not know yet, vs. what we have already known in terms of the subject matter.

Author Response

(The authors gave the same response as above.)

Round 2

Reviewer 1 Report

Reviewer’s Comments

I greatly appreciate the authors' efforts to improve this manuscript. Indeed, it has improved. However, there still remain issues that need to be addressed.

  1. About the Title, “An integrated analysis of VAM and information systems (IS) success model for PropTech service platform. ” It is recommended to change the abbreviation of "vam" to the full name so that readers can understand it at a glance. For example, "An integrated analysis of Value-Based Adoption Model and Information Systems Success Model for PropTech service platform."

  1. About Abstract, on page1, line 8. This paragraph "This study develops a standardized and sustainable PropTech service platform." As suggested by the last review, the term standardization should be used with caution in the article. The reviewer suggests that the author(s) can think about changing it and combining it with the title and text. Author(s) can try to use words such as expansion or integration, verification... to illustrate the research topic want to explore.

  1. About Section 1. Introduction, on page 2, lines 78-79. " In the given context, this study develops a standardized and sustainable PropTech service platform." If the author(s) modify the second point, please also modify this paragraph.

  1. On page 1, line 8. "This study develops a standardized and sustainable PropTech service platform."

On page 2, lines 78-79. "In the given context, this study develops a standardized and sustainable PropTech service platform."

On page 4, lines 164-166." We take a systematic approach to platform design and adopt the  VAM to analyze the perceived value of consumers and the IS success model to standardize the design quality for the IS. "

In these several places, the terms related to the platform may confuse readers. According to the reviewer's point, a platform was not actually developed in this research. The term platform written in the above paragraph should be changed to a pattern or model. It is recommended that the author should think and review the usage of the term "platform" in the article.

  1. About Section 2. Literature review, on page 2, line 92. Suggestion can be changed to "2. Literature review and Hypothesises"

  1. About Section 2.4. Hypothesises, on page 7, lines 308-309. "Figure 1 presents the proposed conceptual model and eleven hypothesized relationships. Please confirm how many hypothetical relationships there are.

  1. About Section 4. Results, on page 10, lines 365-368. Suggest to add relevant literature to explain the value of Gof. Please refer to the following literature.

Tenenhaus, M., Vinzi, V. E., Chatelin, Y. M., & Lauro, C. (2005). PLS path modeling. Computational statistics & data analysis, 48(1), 159-205.

Wetzels, M., Odekerken-Schröder, G., & Van Oppen, C. (2009). Using PLS path modeling for assessing hierarchical construct models: Guidelines and empirical illustration. MIS quarterly, 177-195.

  1. About Section 4. Results, on page 10, lines 365-368. Please add relevant literature to explain the value of SRMR. Suggest to site the following literature.

Benitez, J., Henseler, J., Castillo, A., & Schuberth, F. (2020). How to perform and report an impactful analysis using partial least squares: Guidelines for confirmatory and explanatory IS research. Information & Management, 57(2), 103168.

  1. On page 12, lines 384-385. "In addition, since the correlation coefficient between variables is not large, the possibility of CMB is judged to be small [72]. " Please confirm the word of "CMB". Should it be changed to "CMV"

  1. On page 14, Table 6. Hypothesis testing results. It is suggested that the author(s) can adjust the table to be centered.

Author Response

Dear anonymous reviewer

We thank you for the feedback on our manuscript, “An integrated analysis of Value-Based Adoption Model and Information Systems Success Model for PropTech service platform." We did our best to address every concern in the feedback. The review process significantly improved the quality of our manuscript. We really appreciate it. There are several changes made in the revised manuscript.

It should be noted that we responded to every point from a reviewer, attempted to address it in the revision process, and incorporated it in the revised manuscript. If there are any immediate questions or comments, please do let us know and we will be happy to address them.

Thank you for your consideration. I look forward to hearing from you.

Sincerely,

Jaeyoung Kim

Reviewer 3 Report

I appreciate the author(s) effort to address the reviewers' comments. The paper now reads much stronger and can be a great contribution to the existing body of knowledge.

Author Response

Dear anonymous reviewer

Thank you for your consideration.

Sincerely,